# Recycling of Bovine Hair Waste Through the Design of a Compatibilizing Agent for Sustainable Thermoplastic Starch-Untreated Bovine Hair Composites

**DOI:** 10.3390/polym16233432

**Published:** 2024-12-06

**Authors:** Luz Elena Mora-Maldonado, Anayansi Estrada-Monje, Roberto Zitzumbo-Guzmán, Isis Rodríguez-Sánchez, Leonardo Baldenegro-Pérez, Claudia Ivone Piñón-Balderrama, Saddys Rodríguez-Llamazares, Erasto Armando Zaragoza-Contreras

**Affiliations:** 1Centro de Innovación Aplicada en Tecnologías Competitivas, León 37545, Mexico; lmora.picyt@ciatec.mx (L.E.M.-M.); rzitumbo@ciatec.mx (R.Z.-G.); 2Departamento de Formación Básica Disciplinaria, Unidad Profesional Interdisciplinaria de Ingeniería Campus Guanajuato, Instituto Politécnico Nacional, UPIIG-IPN, Silao de la Victoria 36275, Mexico; isrodriguez@ipn.mx; 3Centro de Ingeniería y Desarrollo Industrial, Santiago de Querétaro 76125, Mexico; leonardo.baldenegro@cidesi.edu.mx; 4Centro de Investigación en Materiales Avanzados, S.C. Miguel de Cervantes No. 180, Complejo Industrial Chihuahua, Chihuahua 31136, Mexico; claudia.pinon@cimav.edu.mx; 5Centro de Investigación de Polímeros Avanzados, Edificio Laboratorio CIPA, Avenida Collao 1202, Concepción 4051381, Chile; s.rodriguez@cipachile.cl

**Keywords:** bovine hair waste, coupling agent, keratin, recycling, sustainable, thermoplastic starch

## Abstract

Bovine hair waste was chemically modified to obtain a coupling agent (CA) for the compatibilization of thermoplastic starch (TPS)-unmodified bovine hair waste (UH) composites. The composites processed with CA presented improved tensile strength (3.5 MPa) compared to TPS–UH composites without CA (1.25 MPa). An interaction mechanism to describe the improvement in mechanical properties and thermal stability was postulated based on Fourier-transform infrared spectroscopy (FTIR) and density functional theory (DFT). In addition, optical and electron microscopy showed that CA favored the adhesion of UH to TPS. Global results suggested the formation of a CA–UH network that interacts with the TPS matrix. Obtaining composites from waste from the tanning industry can contribute to the development of a more responsible and sustainable industry and represents an opportunity to reduce the environmental impact of one of the most important industries globally. It is worth mentioning that this research is aligned with the sustainable development goals (SDGs) proposed by the United Nations, which promotes sustainable industrialization and the promotion of innovation.

## 1. Introduction

The industry of conventional petroleum-derived plastics is continuously expanding due to low production costs and a wide variety of applications [1]. Polymers, known for their flexibility, low weight, excellent processability, and high availability, are widely used by different industries in everyday applications [2,3,4]. Because of the unsustainable nature of petroleum resources and increasing environmental pollution caused by short-life plastics, research on biocomposites from renewable materials has made significant advancements [3,5,6,7].

The global production of keratin wastes is on the rise, with over 6.6 million tons of raw bovine hides and 0.8 million tons of ovine skins being transformed into leather annually [8]. This process generates a significant amount of hair waste, approximately 143,000 tons per year, usually disposed of through dumping, landfilling, and incineration, causing severe environmental damage [9]. Therefore, it is crucial to find eco-friendly methods for hair waste disposal [10,11]. Despite extensive research on hair waste, investigations on bovine hair waste remain commercially underdeveloped [12].

Starch is another important biopolymer often used in creating bio-based blends and composites [13]. It is a renewable resource suitable for industrial applications due to its sustainability. Starch is the most commonly used biodegradable polymer because of its abundance, transparency, non-toxic nature, and excellent oxygen barrier and film-forming properties. Moreover, starch can be obtained using simple techniques, such as extrusion mixing and injection molding. Additionally, its properties can be customized through processing for various applications [14,15]. However, native starch has certain chemical and structural limitations arising from its hydrophilicity, low thermal stability, and the formation of inter- and intramolecular hydrogen bonds that affect its melting properties [16]. Hence, blending it with other components is essential to enhance its properties [17,18].

Keratin has been utilized as a reinforcing agent in composites [19,20,21,22]. However, surface modification of keratin fibers, either by physical or chemical methods, is needed to improve interfacial interaction when mixed with a thermoplastic matrix. As for chemical modification, the hydroxyl groups of keratin facilitate an efficient coupling with the polymer matrix [23]. Studies on keratin-reinforced thermoplastic composites have included Angora rabbit hair [21], wool [24], chicken feather keratin [25,26], keratin from tannery hair [5,27], and human hair waste [28]. A preceding work used bovine hair waste from tanneries (treated with KOH) as a coupling agent in preparing TPS-unmodified bovine hair waste composites processed by injection molding [6]. The authors reported a decrement in the mechanical properties, attributed to excessive processing conditions, reducing the polymeric chain size and tensile strength. This research, on the other hand, examines how processing by compression molding benefits the mechanical properties of the composite. Furthermore, a coupling agent (CA) was added in proportions of 2%, 3%, and 4% to explore the effects of increasing amounts of CA. An improvement in tensile strength and a reduction in water absorption were observed, indicating a better interaction between the components. The latter was evidenced by optical microscopy and SEM.

This research examines how treating bovine hair waste with alkaline affects the thermal and mechanical properties of untreated bovine hair waste–thermoplastic starch composites. These composites were processed using an internal mixing chamber and compression molding. The results indicate that the alkaline treatment improves the interaction between the fibers and the matrix (as determined by FTIR and DFT) and decreases water absorption. This suggests that using alkaline-treated bovine hair waste is a viable way to create sustainable composites.

## 2. Materials and Methods

### 2.1. Materials

Cornstarch (IMSA, Guanajuato, Mexico), raw bovine waste hair (local tannery, León, Guanajuato, Mexico), glycerol (CTR Scientific, Monterrey, Mexico), and potassium hydroxide (Meyer Chemicals) were used as received.

### 2.2. Preparation of Coupling Agent (CA)

UH was washed twice with tap water (controlled at pH 8) in a rotating drum for 30 min and then dried at room temperature. Clean hair was ground to a particle size of about 1 mm and then oven-dried for 24 h at 70 °C. The powder was transferred to a glass vessel and treated with the alkaline solution (0.25 and 0.35 N KOH), keeping the system at 45 °C for 1 h. Then, the dispersion was brought to pH 9 with H_3_PO_4_ and the solid was recovered by filtration. Finally, the product (CA) was dried in an oven at 70 °C for 24 h.

### 2.3. Composite Processing

Composites were prepared as per the data reported in Table 1 and processed in an internal mixing chamber and by compression molding. Thermoplastic starch (TPS) (Figure 1a) was prepared by mixing native cornstarch with glycerol (30 wt%). The untreated hair waste (UH)–thermoplastic starch (TPS) composite (Figure 1b) was prepared by mixing 10 wt% UH with 90 wt% TPS. The rest of the composites, as indicated in the table, contained 10 wt% UH and 2, 3, and 4 wt% hair (treated with KOH 0.25 and 0.35 N as the CA) and the remaining percentage of TPS.

All composites were premixed manually for 5 min and processed in a mixing chamber (CWB, Brabender, Anton Paar, Duisburg, Germany) for 5 min at a temperature profile of 120, 120, and 120 °C and a speed of 80 rpm. Composites were stored in plastic containers for 24 h at room temperature and processed by compression molding in a hydraulic press (Series NE, Carver, Wabash, IN, USA) for 5 min at 145 °C and 20,000 lb.

Mechanical tests were carried out using a universal traction machine (model 3365, series C5057, Instron, Norwood, MA, USA) at 50 mm min^−1^ crosshead speed following the ASTM D638 procedure. The specimen dimensions were according to Type I (Figure 1c), with 3.2 ± 0.4 mm thickness.

### 2.4. Characterization

The structure of composites was investigated using optical microscopy (OM) and SEM. For OM, we used a confocal microscope (VHX-6000, Keyence, Osaka, Japan). Micrographs were obtained from 100× to 2000× for different sections of each sample, and the most representative micrographs were selected. The morphology of the fibers was observed with a scanning electron microscope (JSM-6610LV, JEOL, Tokyo, Japan) in SEI mode with a working distance of 10 mm and operating at 15 kV.

Functional groups were analyzed using a Fourier-transform infrared spectrometer (Nicolet iS10, Thermo Scientific, Waltham, MA, USA). Spectra were acquired in a wavenumber interval from 400 to 4000 cm^−1^, with the attenuated total reflectance (ATR) technique using a diamond tip. Every spectrum was the average of 30 scans.

Vibrational theoretical analysis of the keratin structures around the disulfide bond and the theory of S-S rupture after alkaline treatment was performed using Gaussian 09^®^ [29], evaluating the global minimum with density functional theory (DFT), and using the B3LYP functional and a theory level 6-31+G (d, p) as per the method reported by Mora-Maldonado et al. [6]. In addition, based on the Mulliken scale, analysis of the global and local reactivity descriptors was conducted to determine the local electronegativity of the system atoms and identify interactions between the species [30].

The thermal stability of the raw and treated hair was evaluated using a thermogravimetric analyzer (Q500, TA Instruments, New Castle, DE, USA). The samples were analyzed in a heating ramp from 25 to 600 °C at a heating rate of 5 °C min^−1^ in an air atmosphere. Thermal transitions were determined using a differential scanning calorimeter (DSC PT 1600, Linseis Polyscience, Vielitzerstr, Germany) with a heating rate of 10 °C min^−1^ from 25 to 280 °C under a nitrogen atmosphere at 20 mL min^−1^.

The water absorption test was carried out using the gravimetric method reported by Arpitha et al. with modifications [31]. For this methodology, 10 mm × 10 mm × 3 mm pieces of composite were dried in an oven for 24 h at 70 °C and left to cool at room temperature in a desiccator (*W_i_*). Then, the sample was submerged in deionized water for 24 h and weighed (*W_f_*). Finally, the water absorption percentage (*WA*%) was determined by Equation (1):(1)$$WA\%=\frac{W_{f}-W_{i}}{W_{i}}*100$$

## 3. Results

### 3.1. Mechanical Properties

Figure 2a shows that adding CA to composites containing 10 wt% UH increased Young’s modulus of TPS. The values ranged from 35 MPa for the blank to 178.4 MPa for 4CA25. Young’s moduli were similar for composites containing 2, 3, and 4 wt% CA treated with 0.25 N KOH, with values of 151.1, 170.3, and 178.4 MPa for 2CA25, 3CA25, and 4CA25, respectively. However, for composites with 2, 3, and 4 wt% CA treated with 0.35 N KOH, Young’s moduli were scattered and lower, with values of 99, 136.6, and 177.6 MPa for 2CA35, 3CA35, and 4CA35, respectively. It is worth noting that although Young’s moduli decreased when 2 and 3 wt% CA treated with 0.35 N KOH was added, the values for composites containing 4 wt% were similar, only 0.8 MPa higher than 4CA25.

This behavior was attributed to damage caused to hair by the treatment of a higher concentration of KOH (0.35 N). Furthermore, the combination of alkaline treatment and the increase in the percentage of fiber affected the mechanical properties of the composites. This was due to the saturation of the system added to the fragmentation of polypeptide chains due to alkali treatment, in which KOH 0.25 N was more beneficial than 0.35 N. Muhammad et al. reported that the tensile strength of their biocomposite made with cornstarch reinforced with coconut fiber was reduced in those composites containing a high concentration of fiber, due to its inability to withstand the stress transferred from the polymer matrix [32]. Dąbrowska et al. mentioned that alkaline hydrolysis results in peptides of various sizes depending on the process conditions, such as temperature, time, and alkali concentration, among others. An alkali at high concentration and temperature causes fragmentation, resulting in low-molecular-weight protein fractions, which do not favor the formation of films [33]. On the other hand, Donato and Mija reported that by increasing the percentage of keratin in a mixture, the Young’s modulus decreases and the elongation increases, indicating a plasticizing effect due to low-molecular-weight oligopeptides [34].

In Figure 2b, the addition of CA increased the tensile strength of composites. The blank presented 1.3 MPa, while composites containing CA with 0.25 N KOH were 3.2, 3.6, and 3.5 MPa for 2CA25, 3CA25, and 4CA25, respectively. On the other hand, the tensile strength of composites containing CA with 0.35 N KOH was 3.0, 3.2, and 3.3 MPa for 2CA35, 3CA35, and 4CA35, respectively. Figure 2c indicates that the elongation at break of composites decreased by adding CA, from 26.7 wt% in the blank to 10.9, 9.3, and 9.3 MPa in 2CA25, 3CA25, and 4CA25, respectively. For composites containing CA 0.35 N KOH, the elongation at the break was 12.9, 12.2, and 11.0 MPa for 2CA35, 3CA35, and 4CA35, respectively.

According to a study on natural fiber-based composites, the mechanical properties of such composites are related to the fiber–matrix interfacial bonding [3]. This suggests that CA improves the interfacial bonding of components. Other research conducted on feather keratin–turmeric starch composites revealed that an increase in the percentage of keratin resulted in an improvement in tensile strength and elongation at break. This finding suggests the formation of intermolecular bonds between hydroxyl groups of starch and amino-carboxyl groups of keratin [5]. Flame-retardant biocomposites have also been developed consisting of thermoplastic starch, keratin fibers from tannery hair waste, and aluminum trihydroxide. The study revealed that a low percentage of keratin fiber had a beneficial effect on tensile strength. However, the opposite occurred when a high percentage (30%) of keratin fiber was added. This formulation reduced elongation at break and tenacity, but increased Young’s modulus [26]. Hydrolyzed feather keratin–glycerol composites showed that the addition of glycerol resulted in a decrease in tensile strength and an increase in elongation at break, due to the formation of hydrogen bonds between glycerol and hydrophilic groups of keratin [25]. Also, the addition of grafted coupling agents to polypropylene–wood flour composites increased the modulus of elasticity due to a better intermolecular performance between components. However, the tensile strength decreased because of the poor compatibility between matrix and fiber caused by excessive filler or low coupling agent content [35].

The results of the present study are consistent with these findings. The composites with the highest thermal stability, namely, 2CA35, 3CA35, and 4CA35, showed a decrease in Young’s modulus and tensile strength. Furthermore, hydrogen bonds were formed between -OH groups from starch and glycerol and -NH_2_ and -CO groups from amino acids in keratin, as explained before [35]. The composites containing CA treated with 0.25 N KOH, especially those containing 3 and 4 wt%, resulted in higher values of Young’s modulus and tensile strength. Previous studies indicate that the alkaline hydrolysis of feather keratin with an alkaline treatment produces peptides of different sizes, depending on the treatment conditions (time, temperature, or alkali concentration) [33]. Similarly, this work shows that the higher concentration of KOH did not favor the mechanical properties of composites. Consequently, the treatment with 0.25 N KOH is recommended.

### 3.2. Hair–Matrix Interfacial Interaction

Figure 3 portrays the optical microscopy analysis of composites loaded with UH. The interfacial separation between the fibers and the matrix evidences the lack of compatibility. However, composites containing 2, 3, and 4 wt% CA treated with 0.25 N KOH (2CA25, 3CA25, and 4CA25) exhibited a homogeneous surface and a good interaction between fiber and matrix. The matrix adhered to the fiber’s surface, and deformations were visible due to the strength transmitted between the matrix and hair [36].

It is worth noting that composites containing 2 and 3 wt% CA demonstrate good hair dispersion and adhesion to the polymer matrix [23]. However, when the CA was increased to 4 wt%, the interaction with the matrix was affected and the fibers appeared separated from the matrix with no superficial residues, indicating poor adhesion.

In a related study, increasing the fiber content in Nile rose fiber-reinforced plasticized cornstarch composites resulted in decreased heat transfer, insufficient to complete gelatinization, affecting the adhesion between fiber and matrix and thereby leading to a decay in mechanical properties [37].

In addition to optical microscopy, the interaction among the composite components can be seen in Figure 4. This figure portrays the scanning electron microscopy (SEM) of the blank (composite loaded with untreated hair, UH), 2AC25, and 3AC25 composites.

Figure 4a shows that in the absence of CA, the separation of the fiber and the polymeric matrix is considerable, evidencing weak interfacial interactions. As CA is added, as shown in Figure 4b,c, intermolecular interactions may be formed that promote better interfacial adhesion between the components, resulting in improved mechanical properties, as observed in the previous section.

### 3.3. Functional Group Analysis

The FTIR spectra of blank, 3CA25, and 3CA35 composites are illustrated in Figure 5a–c. As noted, the three samples present a broad band at 3311 cm^−1^ assigned to the O-H group stretching vibration, as reported for keratin-cellulose films [38]. In this region, the formation of hydrogen bonds by adding CA to the composites is expected. The interactions can be explained by the reactive medium given in composites with the cleavage of disulfide bonds, allowing -NH_2_ and C-O to bind. The formation of hydrogen bonds between components containing hydroxyl groups and electronegative compounds was reported in cellulose nanocrystal–polyvinylidene fluoride–perovskite composites [39]. In addition, some shifts between formulations were observed in the absorption bands from 2929 to 2872 cm^−1^, ascribed to the movements when disulfide bonds were broken and hydrogen bonds were formed. A small peak at 1745 cm^−1^ in composite 3CA25 was assigned to the carbonyl group of esters [40]. The band at 1647 cm^−1^ in the blank shifts to 1650 cm^−1^ in the other two composites. Abdullah et al. characterized various starches, assigning a band at 1648 cm^−1^ to the bending of C-O associated with the O-H group [41]. This union corresponds to hydrogen bonds that enhance the composite properties. The small peak at 1560 cm^−1^ in 3CA25 and slight intensity increase in 3CA35 were attributed to the bending of aliphatic and aromatic amines [42,43]. For feather keratin–glycerol composites, the band at 1537 cm^−1^ was assigned to amine II [25]. This shift was ascribed to the formation of hydrogen bonds between N-H groups of keratin and -OH groups of glycerol. The appearance of this small band in composites containing CA may indicate a higher availability of amine groups to form hydrogen bonds with -OH groups of starch and glycerol.

Furthermore, the absorption bands presented by the three composites at 1410, 1018, and 859 cm^−1^ were assigned to the vibration of aromatic rings [42], present in amino acids contained in bovine hair, such as phenylalanine and tyrosine [7,44]. The band at 1018 cm^−1^ is intense and may have overlapped the band at 1020 cm^−1^, assigned to interactions of C-O-C with hydrogen bonds in saccharide films reinforced with keratin [23]. Another signal in the blank appears at 1241 cm^−1^. Similar bands are shown with 3CA25 at 1239 cm^−1^ and 3CA35 at 1238 cm^−1^ with greater intensity. The literature suggests these signals are C-O stretching vibrations [13]. The absorption band at 1144 cm^−1^ was attributed to aromatic alcohols [45]. The three composites in the present work showed a band at 1150 cm^−1^. This may be the product of the interaction between aromatic amino acids and -OH groups of glycerol and starch. The signal at 1078 cm^−1^, assigned to cystine monoxide, an intermediate of cystine oxidation to cysteic acid, is the product of disulfide bond oxidation. Finally, the absorption at 670 cm^−1^ was assigned to C-S and S-S bond extension and C-C bond deformation in keratin-specific sulfur compounds [46]. Table 2 shows the main characteristic infrared absorption bands of the composite materials studied.

According to TGA, complex compounds were formed in composites with the highest content of CA (3CA35), favoring thermal stability. The FTIR spectra show the presence of aromatic amines from amino acids in keratin, which may interact with other functional groups in the composite to form complex compounds and improve thermal stability.

### 3.4. Theoretical Calculations (DFT)

A vibrational analysis was conducted on the interactions between amine-carboxyl, hydroxyl-amine, and hydroxyl-thiol groups, as shown in Table 3. As observed, UH interacts with starch, forming a bond between the amine group (NH) and hydroxyl group (OH) from TPS. UH also interacts with CA through hydrogen bonds between amine (NH) and carbonyl (C=O) groups from amino acids. Furthermore, CA interacts with TPS, forming hydrogen bonds between NH-OH and HS-HO groups. Accordingly, UH presented one site of interaction with TPS. This union was improved by adding CA, which showed two interactions (Table 3). The interactions correspond to the N-H group of CA with a C=O group of UH, and a N-H group of TPS interacting with a -HO group of UH. The alkaline treatment allowed the cleavage of the disulfide bonds (S-S) of keratin, producing a better interaction between the mentioned groups by avoiding steric hindrance. This improvement was reflected in composite properties.

An infrared spectrum was obtained from the DFT study (Figure 6). Table 3 reports the signals assigned to the various interactions between the different materials.

In a previous study, analogous composites were produced, but processed by injection molding, where a TPS–CA–UH-type interaction was proposed [45]. However, in the present work, processing was performed by compression molding. Due to the large differences in shear stresses produced in each type of molding, it is expected that the interactions induced by processing may be different. Consequently, DFT analysis suggested the formation of a network with interactions between CA and UH, and in the middle, the TPS is positioned as illustrated in Figure 7.

Figure 7 represents the arrangement suggested by DFT analysis, in which CA and UH interact first and then extend the interaction to TPS, forming a network that keeps them together, giving rise to a composite with better mechanical performance.

Figure 8 illustrates the arrangement proposed by DFT, in which the distances between the elements with possible interactions were determined. The analysis concluded that the interactions between CA and UH are possible because of breaking the S-S bond. Information on the distances between groups is also included in the figure. The interaction model shows the formation of bonds between UH and the amine group of CA. After this interaction, TPS is positioned between UH and CA, interacting with both of them.

The study of electrostatic interactions allows the approximation of different species without conditioning the union to formal covalent bonds. When the difference in electronegativity is significant, interactions stronger than those of van der Walls are possible. The formation of hydrogen bonds can also be carried out with atoms that are not particularly electronegative, but rather capable of presenting an available concentration of strictly negative charge, as is the case with the S atoms of UH and the CA. Specifically, it is necessary to satisfy two important premises: (1) that a local bond is present and (2) that X-H acts as a proton donor towards A, where A could represent the S atoms. On the other hand, a distance for X-H⋯A where H⋯A of 3.0 to 3.2Å could potentially be considered a hydrogen bond, as is the case for the interaction between 79O-80H and 25S, which presents an approach distance of 2.3086 Å. This not the same for 54O-55H⋯25S, since the distance presented is 4.5594 Å; however, according to the literature, this could be considered a dispersed electrostatic interaction, since the distance exceeds 3.2 Å. On the other hand, the formation of a strong interaction [30,49] on the scale of a hydrogen bond can also be formed between the atoms 22N–24H⋯33O=31C with a distance of 2.1028 Å, catalogued as a possible formation of a hydrogen bond, which allows the effective approach between CA and UH.

Furthermore, according to the vibrational analysis of the structure and bond distances involved, it is observed that these are larger than the normal nature of a structure, and not found in other structures. In this particular case, the O–H bond distance is 0.93 Å, and the analyses for 79O–80H, 54O–55H, and 52O–53H present distances of 1.0009 Å, 1.009 Å, and 0.994 Å, respectively. The bond length reported for N–H is 1.009 Å, and we found 1.0219 Å for 22N–24H and 1.348 Å for 25S–37H, slightly longer than the reported value of 1.34 Å for S-H [30,49]. This suggests that the electrostatic approximation towards electronegatively available atoms in UH and CA is promoted and an approximation between the amine and carboxyl groups of CA and UH. Finally, an electronegativity analysis was performed using the local reactivity descriptor strategy, Fukui functions, and the Mulliken scale, where the local electronegativity values between the atoms of interest were observed.

Approximations 2 and 3 in Table 4 indicate the interactions among the atoms of S, H, and O. As noted, the O–H bonds act as charge donors versus the interaction with the 25S atom that acts as a charge acceptor, presenting positive and negative values, respectively. However, due to the arrangement, the distance of interaction 2 is shorter, thus favoring the formation of a hydrogen bond between the coupling agent and the thermoplastic starch. In the case of approximations 1 and 5, it is observed that the sulfur and oxygen atoms behave as charge acceptors and compete for the interaction with hydrogen between the UH and TPS for approximation 1 and between the UH and CA in approximation 5. Furthermore, interaction 4 is promoted between the amine group of the UH and the carboxyl group of CA where the N–H group acts as a donor and the O of the double bond of the carboxyl group of CA as a charge acceptor. Likewise, the distance observed between these two groups is 2.1028 Å, giving rise to a strong hydrogen bond-type interaction.

### 3.5. Thermal Stability

#### 3.5.1. Thermogravimetric Analysis

TGA was used to investigate the thermal behavior of hairs treated with varying concentrations of potassium hydroxide (KOH). The thermograms of the treated hairs revealed that the first weight loss, between 52 and 86 °C (0.65–1.8%), corresponded to water evaporation. This result is consistent with the findings for yam starch grafted with acrylonitrile [4] and chitosan–starch composites reinforced with keratin from chicken feathers [36]. A second transition appeared in the range of 179 to 194 °C (8–11%), which corresponds to glycerol degradation [50] and coincides with the results for chicken feather keratin–cellulose composites [38]. The third weight loss, between 304 and 307 °C (52–55%), was assigned to a complex process of saccharide ring dehydration, depolymerization, decomposition of keratin [42,51], amylose chain dissociation [29], and glycosidic bond (C-O-C) breakage [4]. The fourth transition occurred between 534 and 578 °C (28–35%), which was attributed to the formation of strong amine bonds (-CH=N) due to cross-linking [38]. The weight loss above 700 °C in the CA-containing composites treated with 0.35 N KOH was attributed to the degradation and oxidation of carbon residues [42]. The weight loss of the composites is shown in Figure 9.

To investigate the effects of increasing the concentration of alkaline treatment, a comparative analysis was performed using the middle percentage of CA, 0.25 N KOH, and 0.35 N KOH added at 3 wt% to the composites, Table 4. The results show that the decomposition temperature increased from 534 °C in the blank to 540 and 579 °C in 3CA25 and 3CA35, respectively. Additionally, the weight loss percentage increased from 6.4% in the blank to 8.4% in 3CA25 but decreased in 3CA35 to 1.5%. In addition, the concentration increment of alkali treatment enhanced the thermal stability of the composites. The literature indicates that improved thermal stability is related to strong amine bond formation (-CH=N) [38]. Furthermore, the amino acid residues in feather keratin–starch–PVA composites form hydrogen bonds through hydroxyl groups, enhancing the molecular size and improving thermal stability [42].

Analysis of the functional groups suggested that complex compounds were formed, especially in composites containing CA treated with 0.35 N KOH, which increases thermal stability. However, such compounds may have an even higher molecular weight in composites with higher concentrations of KOH (0.35 N) and did not seem to bind well with the other composite components. As a result, CA treated with higher alkali concentrations did not improve the mechanical properties of the composites. Therefore, a concentration of 0.25 N KOH is recommended for composite preparation.

Complex compounds may be formed, linked by hydrogen bonds, especially in composites containing CA, which brings about a thermal stability increase. This could be the result of interaction between different functional groups of components, besides processing temperature and alkaline medium. Such compounds will be explained in the functional group analysis below. However, such compounds may have even higher molecular weight in composites with hair treated at higher concentrations of KOH (0.35 N) and do not seem to bind well with the other composite components. Consequently, hair treated with higher concentrations does not improve the mechanical properties of the composites. Thus, it is better to use a concentration of KOH 0.25 N. Table 5 shows a comparative analysis of composites containing untreated hair and the composites with 3% of CA, that is the middle porcentage of CA, to investigate the effects of increasing the concentration of alkaline treatment.

#### 3.5.2. Differential Scanning Calorimetry

The effect of incorporating a CA into the composites is demonstrated in Figure 10a,b. In comparison to composites without CA (63.8 °C), the melting temperature (T_m_) of composites decreased to 59.5, 61.3, and 60.4 °C for 2CA25, 3CA25, and 4CA25, respectively, upon adding CA treated with 0.25 N KOH. Furthermore, CA treated with 0.35 N KOH slightly increased Tm to 64.0, 64.6, and 64.6 °C for 2CA35, 3CA35, and 4CA35, respectively.

The DSC trace of processed composite films made of recycled carbon ashes from biodegradable thermoplastic, maize starch, and plasticized with glycerol, exhibited an endothermic curve upon the melting of the starch crystals. Filler addition resulted in chain mobility in the amorphous starch phase, increasing the glassy region. Crystallization was also altered, and a shift in T_m_ was attributed to an increase in crystallite size rather than a crystallinity modification [52].

We evaluated variations in CA and treatment with different concentrations of alkali. As observed, composites containing CA treated with a higher KOH concentration led to composites with higher melting temperatures, Figure 10b. This phenomenon can be attributed to larger crystals in the composites due to mobility restrictions caused by CA incorporation [52]. The interaction network between the three materials allowed for larger or better-quality crystals, as evidenced by a higher Tm. Furthermore, the higher Tm in composites with CA treated with 0.35 M KOH suggests that the small fractions of CA produce greater movement restrictions in the composites with CA treated with 0.25 M KOH. This alignment of the amylopectin chains results in more ordered and stable crystal structures that require more energy to melt. Proper alignment of the amylopectin chains during the interaction among materials in the composite preparation caused the formation of stabler and more ordered crystal structures that required more energy to melt the crystals.

### 3.6. Water Absorption Assay

The incorporation of CA into TPS significantly affected water absorption, as demonstrated in Figure 11. The composite without CA exhibited a water absorption of 62.7%, while for the composites containing CA treated with 0.25 N KOH, the absorption was 59.6%, 52.4%, and 56.7% for 2CA25, 3CA25, and 4CA25, respectively. Similarly, composites containing CA treated with 0.35 N KOH demonstrated absorption of 51.3%, 48.3%, and 53.9% for 2CA35, 3CA35, and 4CA35, respectively. Notably, composites containing 3 wt% CA showed lower water absorption, with 3CA35 exhibiting the lowest.

Research on keratin–turmeric starch biocomposite films revealed that keratin reduced the solubility of composites due to the strong interaction between polymeric chains within the matrix–fiber system, impeding water molecule penetration. This finding agrees with our observations. In contrast, potato starch–chitosan composites reinforced with keratin feathers exhibit higher water solubility when keratin is treated with NaOH [23]. This can be attributed to the reduction in keratin’s hydrophobic character by alkaline treatment, which is consistent with the results of our study.

It was also observed that composites with lower water absorption had lower Young’s moduli and tensile strength, but exhibited high thermal stability. This may be due to the formation of complex compounds of high molecular weight that reduce compatibility with water [23].

## 4. Conclusions

The growing market for products made of thermoplastic starches with improved properties that satisfy the demands for certain applications has stimulated research on composites fabricated with waste that can be revalued and used in added-value applications. In this research, TPS–unmodified bovine hair waste composites were obtained, which exhibited the formation of an interaction network, CA–UH, that hosted TPS. This produced a composite with improved tensile strength, Young’s modulus, and less water susceptibility. The results may be related to the modification of the crystallization process in the presence of CA. DFT analysis established the distance between the interacting elements. Consequently, an interaction model, UH–CA–TPS, to explain the improvement in the composite mechanical properties was proposed. It was also observed that the material processing had a strong influence on the interaction mechanism among the components and thus on composite mechanical properties. The composite’s biodegradability characteristic and the water resistance improvement make this composite suitable for application as packaging materials, flowerpots, or sprouters for plants, as it has high carbon and nitrogen content from the bovine hair, which could provide useful nutrients.

## Figures and Tables

**Figure 1 polymers-16-03432-f001:**
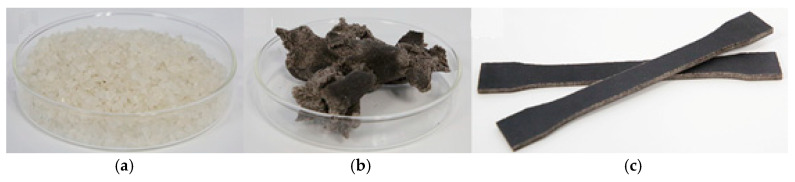
Images of (**a**) thermoplastic starch (TPS), (**b**) blended composite, and (**c**) test specimens.

**Figure 2 polymers-16-03432-f002:**
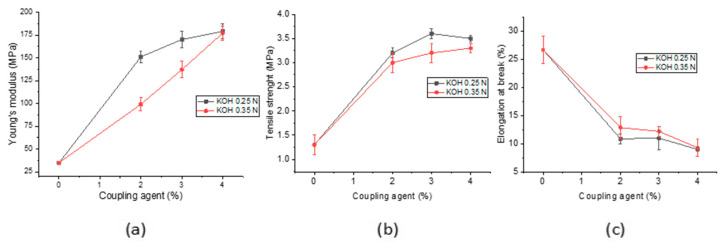
Effect of percentage and concentration of CA on (**a**) Young’s modulus, (**b**) tensile strength, and (**c**) elongation at break of the composites.

**Figure 3 polymers-16-03432-f003:**
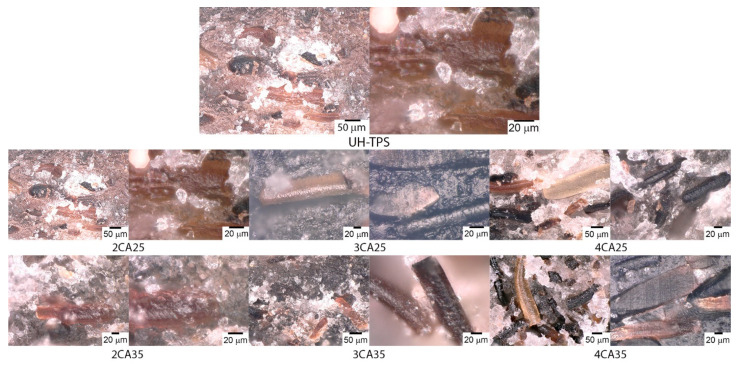
Optical microscopy analysis of composites.

**Figure 4 polymers-16-03432-f004:**
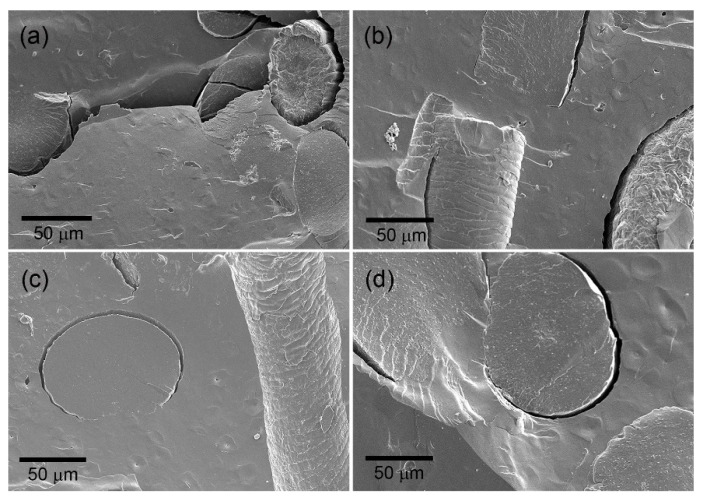
SEM of (**a**) composite loaded with untreated hair, (**b**) 2CA25, (**c**) 3CA25, and (**d**) 4CA25.

**Figure 5 polymers-16-03432-f005:**
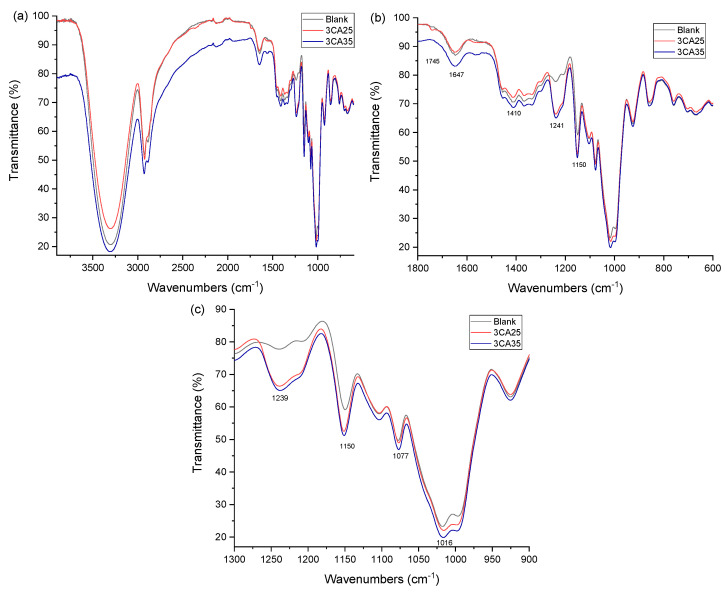
FTIR spectra of composites analyzed. Black = blank, blue = 3CA35, and red = 3CA25. (**a**) Infrared in the wavenumber interval of 3600 to 1000 cm^−1^, (**b**) Infrared in the wavenumber interval of 1800 to 600 cm^−1^ and (**c**) Infrared in the wavenumber interval of 1300 to 900 cm^−1^.

**Figure 6 polymers-16-03432-f006:**
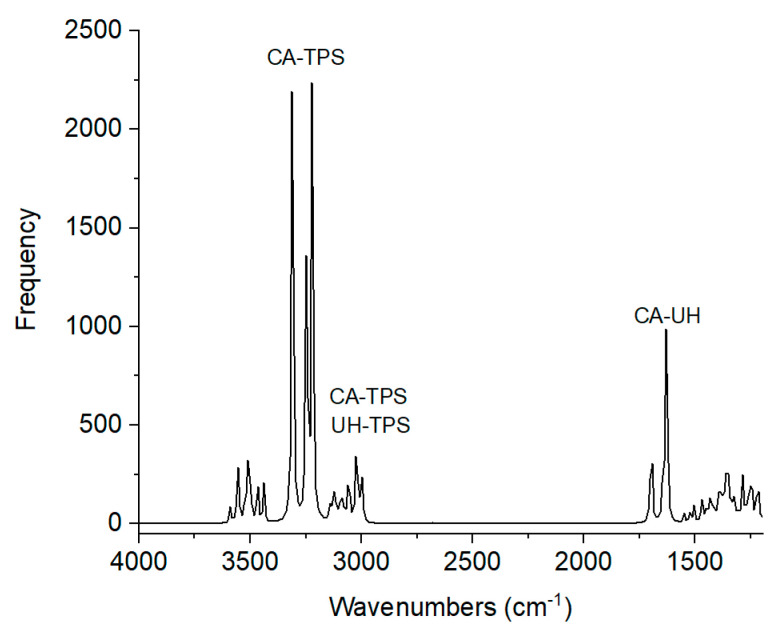
Infrared spectrum obtained from DFT analysis.

**Figure 7 polymers-16-03432-f007:**
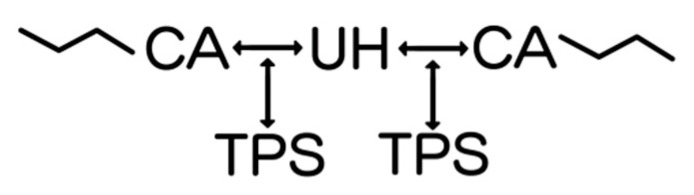
The network formed by the CA–UH–TPS interaction.

**Figure 8 polymers-16-03432-f008:**
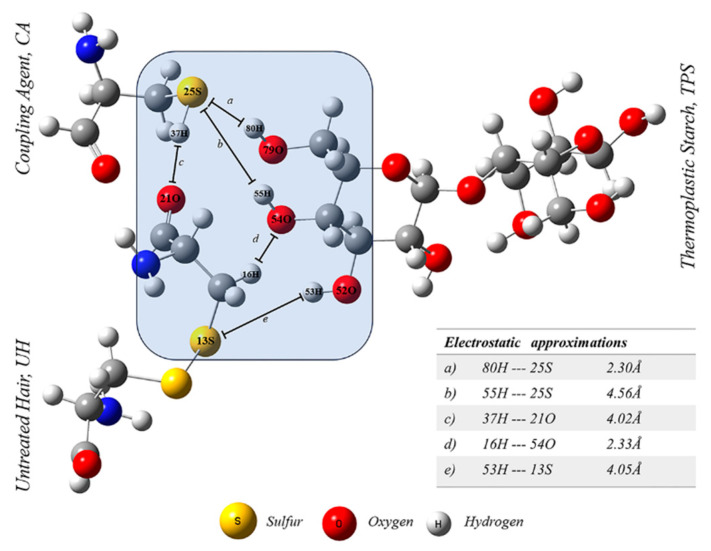
Model of interaction among the components of the composite material.

**Figure 9 polymers-16-03432-f009:**
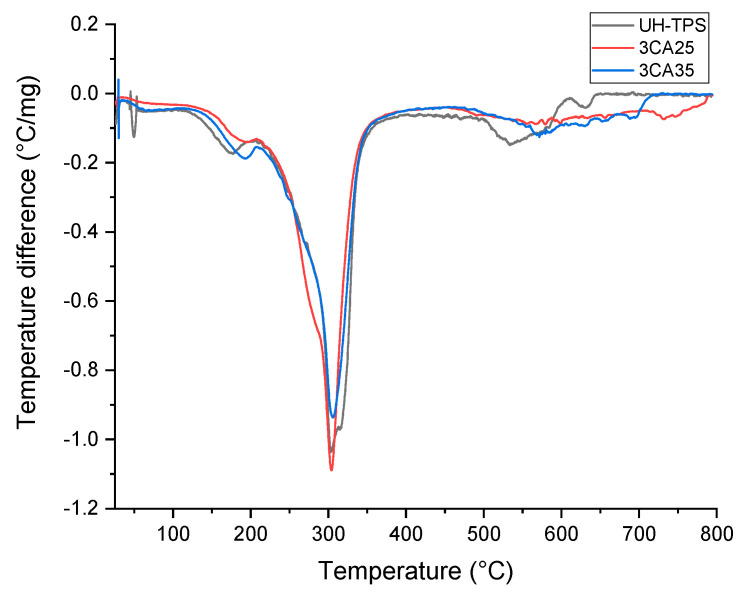
TGA traces of composites: UH-TPS, 3CA25, and 3CA35.

**Figure 10 polymers-16-03432-f010:**
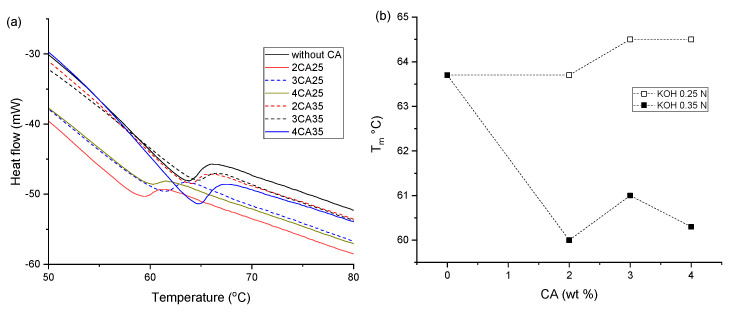
(**a**) DSC thermograms of composite materials and (**b**) T_m_ vs. CA in the composites.

**Figure 11 polymers-16-03432-f011:**
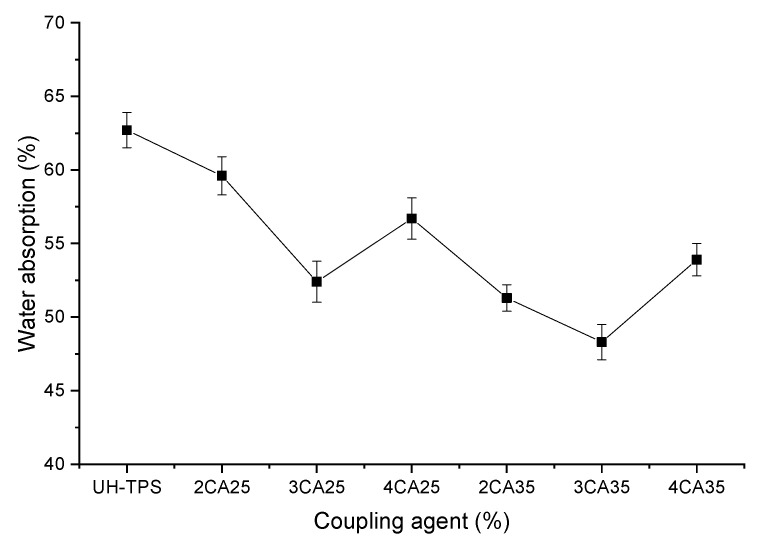
Water absorption of composites.

**Table 1 polymers-16-03432-t001:** Composition of composites.

**Composites**	**UH (wt%)**	**TPS (wt%)**	**CA (wt%)** **Treated Hair, KOH 0.25 N**
TPS	0	100	0
UH-TPS	10	90	0
2CA25	10	88	2
3CA25	10	87	3
4CA25	10	86	4
			**CA (wt%)** **Treated Hair, KOH 0.35 N**
2CA35	10	88	2
3CA35	10	87	3
4CA35	10	86	4

UH: untreated hair waste; TPS: thermoplastic starch; CA: coupling agent.

**Table 2 polymers-16-03432-t002:** Band assignation in the FTIR spectra.

	Wavenumbers (cm^−1^)				
	Experimental Values			Values Reported in Literature	References
Bonding	UH-TPS	3CA25	3CA35		
O-H stretching	3301	3311	3310	3300–3400	[38]
Asymmetric/symmetric CH_2_/CH_3_	2928/2888	2925/2882	2929/2886	2929/2872	[13]
Stretching C=O	-	1745	-	1744	[40]
C-O bending associated with OH group	1647	1650	1650	1650–1800	[41]
NH bending vibration/amine	-	1566	1566	1545/1574	[42,43]
Vibration of aromatic rings	1410	1412	1411	1410, 1018, and 865	[42]
Aromatic alcohols -OH	1150	1150	1151	1144	[45]
Cystine monoxide -S-SO/	1078	1077	1077	1078	[44]
Stretching C-O	1106	1106	1106	1102	[47]
Vibration of aromatic rings/interactions of C–O–C bond stretching in saccharide films with hydrogen bonds	1018	1016	1016	1018	[42]
stretching vibration C-O	925	925	925	964	[42]
Vibration of aromatic rings	857	859	859	865	[42]
Vibrations of the–CH=CH– bonds of benzene rings	759	759	759	743, 759	[48]
Extension of C-S and S-S bonds	672	667	667	670	[46]

**Table 3 polymers-16-03432-t003:** Interactions among functional groups of UH, CA, and TPS.

Component of Composite	Interactions	Absorption (cm^−1^)	Absorption
CA–UH	N-H-------O=C	1624, 1627	Symmetrical and asymmetrical stretching
TPS–UH	N-H-------O-H	3449, 3553, 3565,	Symmetrical stretching and twisting

**Table 4 polymers-16-03432-t004:** Analysis of electrostatic approximations.

No. of Approximation	Interaction	Atom	Local Electronegativity (eV)	Interaction Distance (Å)	Interaction Type
1	13S⋯53H–52O	13S	−0.01030	4.0569	Dispersed electrostatic interaction
		53H	0.00057		
		52O	−0.00075		
2	25S⋯ 80H–79O	25S	−0.00754	2.3086	Potential hydrogen bond
		80H	0.00006		
		79O	0.00094		
3	25S⋯55H–54O	25S	−0.00754	4.5594	Dispersed electrostatic interaction
		55H	0.00058		
		54O	0.00140		
4	22N–24H⋯33O	22N	−0.00630	2.1028	Potential hydrogen bond
		24H	−0.00290		
		33O	0.00153		
5	25S–37H⋯ 21O	25S	−0.00754	4.0247	Dispersed electrostatic interaction
		37H	0.00071		
		21O	−0.00764		

**Table 5 polymers-16-03432-t005:** Comparative analysis of composites. Composite containing untreated hair and medium treatment composites. Blank (UH-TPS), 3CA25, and 3CA35.

UH-TPS		3CA25		3CA35	
T (°C)	Weight Loss (%)	T (°C)	Weight Loss (%)	T (°C)	Weight Loss (%)
52.2	1.5	72.5	4.3	72.5	4.0
179.2	10.1	195.9	7.6	193.9	7.8
304.5	41.9	306.3	44.4	304.6	42.1
534.6	40.2	540.3	-	578.8	-
632.3	6.3	667.3	-	597.2	-

## Data Availability

The original contributions presented in the study are included in the article, further inquiries can be directed to the corresponding author.

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
