# Peer review of "Recycling of Bovine Hair Waste Through the Design of a Compatibilizing Agent for Sustainable Thermoplastic Starch-Untreated Bovine Hair Composites"

_polymers, 2024, doi:10.3390/polym16233432_

Round 1
Reviewer 1 Report
Comments and Suggestions for Authors
The article is devoted to the study of obtaining and using a coupling agent (CA) for composites based on thermoplastic starch with bovine hair. In general, this is an interesting work with a relevant environmental theme. The authors used a well-thought-out approach, a large number of analytical research methods and modeling. The authors also paid sufficient attention to the analysis of literary data on similar composites and comparison of their data with them. Nevertheless, there are a number of questions to this manuscript, the solution of which will make it better:
1. Did the obtained CA particles stick together, agglomerate? If so, how was this avoided? How was the uniformity of distribution of both CA and UH in the mixture controlled? The authors should add information about this.
2. During the IR studies, were the experiments carried out at different points of the sample? How well did these data correlate with each other? According to Fig. 3 (view of the samples from an optical microscope), the samples are highly heterogeneous. With what aperture were the IR spectra then recorded? How many points of one sample were analyzed? How were the IR spectra processed after receipt before analysis?
3. In Fig. 2 there are no data on the measurement error, which is especially important when interpreting the data in Fig. 2b and 2c. In general, these figures show that the introduction of CA has a good effect on the composite, but it is surprising that the amount of KOH has a strong effect on a small introduction of CA (2-3%), and has virtually no effect on 4%. Either there is a large measurement error here, within which the points for different amounts of KOH fit, or the authors should explain why KOH has almost no effect on a high concentration of CA in the composite. Still, the difference between the amount of introduced CA is too small (+-1%) to talk about a greater influence of the interactions that arise. This is especially strange in the context of the fact that below (lines 197-200) the authors write that when going to 4% CA, the adhesion of the fibers becomes poor.
4. Why are the SEM data for 4% CA not shown in Fig. 4? Since the authors above talk about deterioration of adhesion for such a sample, it would be interesting to see this in the SEM data.
5. The caption to Fig. 5 indicates a "green" spectrum, although it is not shown in the figure. And unfortunately, not all peaks analyzed in the text are shown in this figure. For example, the peak at 1745 cm-1 in the red spectrum is not shown in any way and is so small that it is difficult to see with this scale of the spectra. Perhaps the authors should somehow supplement the figure with additional inserts with enlargements of individual sections of the spectrum that they are talking about.
6. There are no IR and TGA data for the sample with 2 and 4% CA. Why?
7. The data in Fig. 10a are very poorly visible. There are no data on the measurement error in Fig. 10b.
8. Fig. 11 - no indication of the measurement error.
Minor comments:
-In Figure 2, the font size is very small and should be changed.
-Line 183 – a reference should be added after the words “…as explained before”.
-The scale grid in Figure 3 is so small that it is not visible.
-Line 281 indicates Figure 6. Is this correct? It looks more like Figure 7 in meaning.
-In Figure 8, it is indicated that the yellow balls are sulfur, the red ones are oxygen, and the gray ones are hydrogen. And what do the blue balls show? Nitrogen? This should be indicated in the figure.
- Authors need to work on formatting the reference list according to the journal's requirements.
Author Response
Attached you will find a document with the answers to your comments.
Thanks

Reviewer 2 Report
Comments and Suggestions for Authors
The manuscript “Recycling of bovine hair waste through the design of a compatibilizing agent for sustainable thermoplastic starch composites” by Anayansi Estrada-Monje et al. deals with the development of sustainable thermoplastic starch composites containing bovine hair waste both as coupling agent and unmodified component. The manuscript is interesting and contributes to the sustainable industrialization and the promotion of innovation. This study was conducted on commercially thermoplastic starch (TPS) mixed with glycerol, 10 wt% unmodified bovine hair (UH) and 2-4 wt% UH treated with solution of 0.25 N KOH and 0.35 N KOH, as coupling agent (CA). The proposed tested methodology involved the measurement of tensile properties, ATR-FTIR analysis, optical and electronic microscopy, thermal properties (TGA and DSC), and water analysis. A model for arrangement of the composite components by DFT was proposed.
For improving the manuscript, I suggest the following additions and revisions.
1) From the actual title, it is understandable that bovine hair waste was used only for the preparation of a coupling agent. I proposed the modification of title as “Recycling of bovine hair waste through the design of a compatibilizing agent for sustainable thermoplastic starch-untreated bovine hair composites”. “Thermoplastic starch” could be added to the keywords.
2) The Introduction Section is not well organized. It should be restructured to highlight the significant advantages of composites containing keratin according to literature; the authors should explain which is the advancement of this work compared to existing studies and clarify why 2-4 wt% CA were used in the composites?
3) Line 53: please mention what are “simple techniques” for obtaining of starch.
4) Please add more references for this phrase: “Keratin has been utilized as a reinforcing agent in composites [17]”.
5) Please check the accuracy of this phase: “As for chemical modification, the hydroxyl groups of keratin facilitate an efficient coupling with the polymer matrix [18]”.
6) Please specify at the 2.2 Section the ratio between UH and KOH solution.
7) Please specify whether the raw bovine waste hair (UH) was used as received for preparing the starch composites. Did the authors consider that the dimension of fiber could influence their interaction with the matrix?
8) Line 114: wavelength should be replaced with “wavenumber”.
9) Optical microscopy is missing from the Characterization Section. The order of the measurements from the Result Section tests does not align with that from the 2.4 Section.
10) The results did not show the standard deviation. It was expected that ten specimens were tested for tensile properties measurement.
11) Figure 3 does not include a clear scale bar of images, making it difficult to interpret. It is not well understanding what means “Untreated hair” from first subfigure. Does it refer to untreated hair before processing? Please use the same codification as shown in table 1 (for example, UH-TPS).
12) I suggest to use “Blend” or “composite” throughout the manuscript.
13) 3.2. The Hair-matrix interfacial interaction Section needs revision. In Lines 193-195, the authors affirmed that “composites containing 2, 3, and 4 wt% CA treated with 0.25 N KOH (2CA25, 3CA25, and 4CA25) exhibit a homogeneous surface and a good interaction between fiber and matrix”, while it lines L98-200, appeared that “when CA increases to 4 wt%, the interaction with the matrix is affected, and the fiber appears separated from the matrix with no superficial residues, indicating poor adhesion”. Please clarify.
Untreated hair was used in all composites, only CA contains treated hair. The caption of Figure 4 should be updated accordingly.
14) Line 268: “This union is improved by adding CA, which shows three interaction sites, Table 3”. Please clarify.
15) Conclusion Section needs revision. Based on the obtained results, can the authors recommend specific applications for these composites? Additionally, specify at which SDG could this work contribute to. Please use the same codification/abbreviation along all manuscript (for example “TPS/keratin composites” or “TPS/unmodified bovine hair waste”).
Author Response

(The authors gave the same response as above.)

Reviewer 3 Report
Comments and Suggestions for Authors
It is sound both surprising and prospective to modify bovine hair to obtain a coupling agent for composite.
Obtaining composites from waste from the tanning industry. It sounds fine.
For these reasons, the paper may be published. Of course, it is a discursive paper, not a paper finalizing researchers. I can recommend this paper for publication as a discursive paper. If the authors want to formulate a final result, please, do it carefully and justify the result.
I do not find any marks of Density Functional Theory, as I know this. I strongly recommend clarifying the point(s) related to the use of Density Functional Theory.
Author Response

(The authors gave the same response as above.)

Round 2
Reviewer 2 Report
Comments and Suggestions for Authors
The manuscript was improved compared with the previous form. A new valuable explanation about the electrostatic interactions was added. The revision of authors is satisfactory for me in this form.
Minor corrections/suggestion are provided below:
Instead of the new phrase “Authors have dedicated their studies to sustainable polymers such as cellulose, starch, lignin, collagen, and keratin” I recommend the authors to insert their studies as references in previous phrase [3,5].
Line 72: coupling agent (CA)
Why the authors did not mention in Material Section the pre-treatment of UH and size smaller than 0.1 mm?
Figure 3 contains notation AC. Please modify it with CA. Idem for Figure 5, Figure 10 (a), and Figure 11. Idem at the caption of Figure 4.
The meaning of some phrases should be checked. For example: “Table 4 reports approximations 2 and 3”.
I suggest the authors to use in manuscript the explanation from cover letter with tensile tests (especially Dąbrowska et al. and Donato and Mija) after the Figure 2.
Author Response
Thank you for your review. We have responded to Reviewer 2's comments, attached a document with the aswers, and highlighted the changes in the manuscript in color.
